# Basic Principles in the Design of Spider Silk Fibers

**DOI:** 10.3390/molecules26061794

**Published:** 2021-03-23

**Authors:** José Pérez-Rigueiro, Manuel Elices, Gustavo R. Plaza, Gustavo V. Guinea

**Affiliations:** 1Centro de Tecnología Biomédica, Universidad Politécnica de Madrid, 28223 Madrid, Spain; m.elices@upm.es (M.E.); gustavoramon.plaza@ctb.upm.es (G.R.P.); gustavovictor.guinea@ctb.upm.es (G.V.G.); 2Departamento de Ciencia de Materiales, ETSI Caminos, Canales y Puertos, Universidad Politécnica de Madrid, 28040 Madrid, Spain; 3Biomedical Research Networking Center in Bioengineering, Biomaterials and Nanomedicine (CIBER-BBN), 28029 Madrid, Spain

**Keywords:** spider silk, mechanical properties, X ray diffraction, spider silk standardization initiative (S3I)

## Abstract

The prominence of spider silk as a hallmark in biomimetics relies not only on its unrivalled mechanical properties, but also on how these properties are the result of a set of original design principles. In this sense, the study of spider silk summarizes most of the main topics relevant to the field and, consequently, offers a nice example on how these topics could be considered in other biomimetic systems. This review is intended to present a selection of some of the essential design principles that underlie the singular microstructure of major ampullate gland silk, as well as to show how the interplay between them leads to the outstanding tensile behavior of spider silk. Following this rationale, the mechanical behavior of the material is analyzed in detail and connected with its main microstructural features, specifically with those derived from the semicrystalline organization of the fibers. Establishing the relationship between mechanical properties and microstructure in spider silk not only offers a vivid image of the paths explored by nature in the search for high performance materials, but is also a valuable guide for the development of new artificial fibers inspired in their natural counterparts.

## 1. The Singular Tensile Behavior of Spider Silk Fibers

The essence of materials science is clearly exposed when facing the challenge of creating a material with a given set of properties, and considering how these properties could arise from a defined microstructure. In most cases, the interest is driven by the need of generating some singular features that differ from those exhibited by existing materials. Consequently, these singular features are supposed to result from some singular microstructural features. As described in this review, the study of spider silk offers a nice example of how these guiding ideas may be applied in practice. In addition, the natural origin of spider silk implies that a thorough understanding of the material requires unveiling the principles used by nature in the production of these fibers [1,2]. Those principles might be used for the production of novel artificial fibers inspired in their natural counterparts within the framework of the emerging area of biomimetics [3].

The unique tensile properties of spider silk were recognized from the earliest works in the field [4,5] as the result of combining unusual values of both tensile strength and strain at breaking [6,7,8]. This combination, in turn, leads to values of the work to fracture (the energy required to break the material which is measured as the area below the stress-strain curve of a material) in excess of 200 MJ/m^3^, much larger than other high performance fibers, such as Kevlar [9]. Simultaneously, a much less desired property of the material was also recognized: its extreme variability in terms of mechanical behavior [10,11]. Although it is assumed that this variability may represent an evolutionary advantage, since it allows to adapt the properties of the fiber to the immediate requirements of the spider [11], it was soon realized that this feature represented a fundamental drawback for the thorough characterization of the material. Thus, many pressing questions related with the extent of this variability remained largely unanswered [12,13,14], most importantly those related with the differences in the silks spun by different spider species [15].

This variability is illustrated in Figure 1a, where a set of representative stress-strain curves of major ampullate spider silk (MAS) fibers spun by *Argiope trifasciata* specimens is shown. Major ampullate gland silk is employed in the building of the web, and it makes the lifeline spun by the spider while crawling or climbing [16]. Its role in the structural integrity of such critical elements for the survival of the animal led to its consideration, since the initial works in the field, as the most optimized fiber produced by the spider in terms of its mechanical performance [17]. As a matter of fact, the proportion of studies on major ampullate gland silk compared to those on other spider silk types is so high that the term “spider silk” is mostly used as a synonym of “major ampullate gland silk”. This convention will be followed in this review, except when indicated explicitly that the results are referred to a different spider silk type.

Figure 1a compares the tensile properties of *Argiope trifasciata* spider silk fibers obtained from three different origins: forcibly silked (FS), naturally spun (NS) and maximum supercontracted (MS). FS fibers are obtained by attaching the fiber to a rotating mandrel and pulling, in a process known as forced silking [18]. Natural spun fibers are retrieved either from the web [19] or from the lifeline of a crawling or climbing spider [20]. Lastly, maximum supercontracted fibers are obtained by allowing either FS or NS fibers to supercontract. At this moment, it may be worth giving to the concept of supercontraction its due relevance for the study of spider silk.

Supercontraction was first recognized as a significant reduction of the length of spider silk fibers when immersed in water or in environments with high relative humidity [21], and was later critical to understand the importance of the elastomeric behavior in defining the tensile behavior of spider silk [22,23]. Initially, it was hypothesized that the biological function of supercontraction might be related with the recovery of a web after being deformed [24]. The discovery of supercontraction in silk fibers spun by other spiders of the Entelegynae, but not belonging to the Orbiculariae (orb-web weaving spiders) group [25,26], however, showed that this phenomenon might play a central role in the design of silk fibers. In this regard, it was found that the maximum supercontracted state constitutes a ground state of the material, to which the fiber may return independently of its previous loading history by being immersed in water [27]. The existence of this ground state also allowed understanding the origin of the variability observed in the tensile properties of the fibers spun by a given species as simply reflecting the stretching of a given fiber with respect to the ground state [28]. A basic consequence of this finding was the possibility of classifying the different true stress-true strain curves using an alignment parameter, α, defined from the length of a given fiber, *L*, divided by the length of the fiber after being subjected to maximum supercontraction, *L_MS_*, as [28,29]:(1)$$\alpha =Ln\left(L/L_{MS}\right)$$

Figure 1b shows the stress-strain curves of fibers spun by *Argiope argentata* spiders and illustrates how the variability observed initially when characterizing spider silk fibers is reduced basically to the level of the precision allowed by the experimental procedure when the fibers are subjected to maximum supercontraction. It should also be noticed that the tensile properties of the fibers in Figure 1 are expressed as true stress-true strain curves. Tensile testing the fibers requires the measurement of the force exerted on the fiber and of the increase of length corresponding to that force. These results are conventionally expressed as stress, which is obtained by dividing the force by the cross sectional area, and as strain, which is obtained by dividing the increase in length by the length of the fiber. Conventionally, there are two possibilities for defining stress and strain following these definitions. Thus, engineering stress, *s*, is defined as force, *F*, divided by the initial cross sectional area of the sample, *A*_0_.
(2)$$s=\frac{F}{A_{0}}$$
and engineering strain, *e*, is defined as the increment in length, Δ*L*, divided by the initial length of the fiber, *L*_0_.
(3)$$e=\frac{\Delta L}{L_{0}}$$

Alternatively, it is possibly to define the true stress, *σ*, as force divided by the instantaneous area, *A*.
(4)$$\sigma =\frac{F}{A}$$
and true strain, *ε*, as the increment in length, *dL*, divided by the instantaneous length, *L*.
(5)$$\varepsilon =\frac{dL}{L}$$

Both engineering strain and true strain contain exactly the same information and it is possible to convert one magnitude into the other with the following relation.
(6)ε=Ln(1+e)

The interconversion between true stress and engineering stress requires and additional hypothesis on how the cross sectional area of the material varies upon deformation. If the volume of the material remains constant (a hypothesis whose validity was established for spider silk [30]) the following relationship applies:(7)σ=s(1+e)

In most materials the choice between engineering and true magnitudes is inconsequential, since both sets of magnitudes yield comparable values for low values of strain. However, large differences between the numerical values of both sets appear for larger strains. In this regard, the large values of strain attained by spider silk fibers suggests that the analysis of its tensile properties can be simplified by using true instead of engineering magnitudes, as indicated in Figure 1.

As illustrated in Figure 1b, the existence of a ground state that can be reached by allowing the fiber to supercontract fully offers a convenient way to remove the variability in the mechanical properties of fibers spun by a given species. However, large differences are still found when comparing the true stress-true strain curves of fibers spun by different species (Figure 2a), even if fibers are compared after being subjected to maximum supercontraction. In this context, it was realized that one of the consequences of the definition of the alignment parameter, *α*, was the concurrence of all fibers spun by a given species at high values of strain [29], and it was checked if a similar concurrence might be found when comparing true stress-true strain curves of fibers spun by different species after being subjected to maximum supercontraction. The result of this procedure is illustrated in Figure 2b, where it is apparent that all true stress-true strain curves do concur at high values of strain with the maximum supercontracted fiber spun by *Argiope aurantia* spiders, which was taken as the reference curve [29]. 

The usage of this procedure allows classifying the true stress-true strain curve of any given species with a single parameter, α* (alpha-star), that is defined as the displacement along the true strain axis (X-axis) of the sample curve required to concur with the reference curve at high values of strain (see Figure 2b). The availability of the α* parameters of a wide range of spider species would help establish correlations between the tensile behavior of the material and the evolutionary history and/or the ecological niche occupied by a given species. This information, in turn, would be applicable in areas such as evolutionary biology and material science, and its compilation has thrust the spider silk standardization initiative (S3I. www.ctb.upm.es/core-facilities, accessed on 22 March 2021) [31] for the determination of the α* parameter worldwide in a set of spider species as wide as possible. The possibility of defining the α* parameter suggests that Entelegynae spiders might follow a common design pattern for the spinning of their fibers. The unveiling of this pattern, however, requires the acquisition of a thorough set of microstructural data on this material. 

## 2. The Semicrystalline Organization of Spider Silk

The origin of the singular properties of spider silk can be traced down to the essential characteristics of their constitutive biomolecules, that include genetic organization [32], amino acid selection [33] and arrangement of the amino acids into protein sequences [34,35]. Not surprisingly, the formation of the fibers do depend on the assembly of the proteins, following a master plan established at these three levels [36]. In particular, all silk fibers are shown to contain two phases: crystalline and noncrystalline, that yield what is described as a semicrystalline microstructure.

The semicrystalline microstructure of spider (and silkworm) silk was established from the earliest works in the field [37,38,39]. In both silks there appears a nanocrystalline phase that results from the piling up of β-sheets. The β-sheets of spider silk are formed from the polyalanine (–A_n_–) motifs [34], while those of *Bombyx mori* silkworm silk correspond to the –GAGAGS– motifs [40]. In spite of using such powerful microstructural characterization techniques as nuclear magnetic resonance [41], the present knowledge on the non-crystalline phase (sometimes referred to as amorphous phase) of silks is much more scarce, although it was possible to identify various secondary structure motifs in this phase [42,43], as well as the relationship between these motifs and the appearance of supercontraction [44,45].

The microstructural characterization of the crystalline phase of spider silk took advantage of using synchroton sources for the characterization of single fibers [46]. In particular, the combination of single fiber X ray diffraction and the possibility of harnessing the variability of the fibers through supercontraction allowed a thorough comparison of the crystalline phases of a set of representatives from the Entelegynae suborder [47]. Figure 3a shows the X-ray diffraction of silk fibers spun by representatives of the two subgroups that form this suborder: RTA-clade and Orbicularians. All diffraction patterns were taken from individual fibers of the corresponding species after being subjected to maximum supercontraction. The analysis of the diffraction patterns showed that all analysed species of the Entelegynae presented a common unit cell, which corresponds to the β(3) group of the classification proposed by Warwicker [38].

In addition to identifying the unit cell of a crystalline phase, X ray diffraction also allows extracting information on three extremely relevant microstructural features of silk fibers: the size of the nanocrystals, the orientation of the nanocrystals with respect to the macroscopic axis of the fiber, and the ratio between crystalline and amorphous phases, usually measured as the percentage of crystalline phase or crystallinity. Thus, establishing the similarities and differences in these microstructural features among the fibers spun by different species would be a valuable hint to determine their likely influence on the variability observed among the studied species. The analysis of the size of the nanocrystals did not show remarkable variations between the fibers and, consequently, this microstructural feature was not considered relevant in causing the observed differences. In contrast, significant differences were observed when analyzing the orientation of the nanocrystals with respect to the macroscopic axis of the fiber (measured as the full width at half maximum of the diffraction spot along the azimuthal direction). A similar situation was found to apply when comparing the values of crystallinity. In particular, a clear correlation was found between the percentage of supercontration (%SC = *L*/*L_MS_* − 1, where *L* is the original length of the fiber, and *L_MS_* is the length of the fibers after maximum supercontraction) and both microstructural features, a trend that was especially clear when the orientation of the nanocrystals was considered (Figure 3b,c).

## 3. Identifying the Microdeformation Mechanisms in Spider Silk

The application of a similar strategy based on the characterization by X ray diffraction of individual fibers of known tensile behavior to silk fibers spun by a single species, but with different values of the alignment parameter, *α*, allows analyzing the dynamics of these microstructural features upon deformation of the material. Figure 4a shows the diffraction patterns obtained from spider silk spun by *Argiope trifasciata* spiders at different values of the alignment parameter, and indicate the position of those parameters along the true stress-true strain curve of a maximum supercontracted fiber [48].

In parallel with the analysis on the fibers spun by different entelegynae species, the X ray patterns were employed to extract information on the orientation of the β-nanocrystals with respect to the macroscopic axis of the fiber (measured as FWHM), the crystallinity, and the size of the β-nanocrystals (Figure 4b–d). Thus, it was found that changes in the orientation of the β-nanocrystals and the crystallinity proceeded sequentially: the orientation of the β-nanocrystals increased up to a value of the alignment parameter of α = 0.4, and remained constant from that value on. In contrast, the crystallinity remained constant up to the same value of the alignment parameter of α = 0.4, and increased monotonously from that value on. No variation was recorded on the size of the nanocrystals in any of the three spatial directions, consistently with previous works [49].

The evolution of this set of microstructural features was compatible with the existence of two regimes in the deformation of silk fibers outside the initial elastic regime [50,51]. Firstly, the β-nanocrystals do rotate and become aligned with the macroscopic axis of the fiber. Secondly, the proportion of the crystalline phase increases at the expense of the amorphous phase. No clear hints, however, could be found that justified the appearance of the new crystalline regions. The most obvious hypothesis, which implied the growth of the pre-existing β-nanocrystals was apparently precluded by the constancy observed on their size (Figure 4d). An alternative mechanism that assumed the creation of new poly-alanine nanocrystals not related to the original set would imply that some poly-alanine sequences in the amorphous phase should assemble to yield the new crystalline phase. A possible solution to this contradictory results was provided by extending the study at other spider silks different from that produced in the major ampullate gland.

Beside major ampullate gland silk, Orbicularian spiders spin other silks with differing compositions, microstructures and properties, whose comparison with major ampullate gland silk casts light on the design of these materials. Minor ampullate gland silk is often obtained during the forced silking process accompanying major ampullate gland silk fibers and, although it shares a semicrystalline microstructure with them, it does not show supercontraction [52]. In contrast, the silk produced in the flagelliform gland, that constitutes [53,54] the other major fibrillary element that builds up the prototypical bidimensional webs characteristic of the Orbicularians [12] do show supercontraction and is characterized by the existence of a ground state. In comparison with the wide range of studies devoted to the characterization of major ampullate gland silk, the knowledge on flagelliform silk has lagged mostly due to two basic practical difficulties. To begin with, flagelliform silk can be only obtained in small quantities by direct retrieval from the web, since it is not possible to do any forced silking with these fibers. In addition, the presence of the gluey solution produced in the aggregate gland [55,56] that covers the flagelliform fiber largely hampers both the mechanical and microstructural analysis of this material.

In spite of these drawbacks, it was found that flagelliform silk also shows a ground state to which the fiber may return through supercontraction [57]. Consequently, the analysis of the evolution of flagelliform silk from its ground state is a valuable complement to the previous data on major ampullate gland silk. The comparison between major ampullate and flagelliform gland silks is even more relevant since flagelliform gland silk lacks the poly-alanine motifs that form the β-polyalanine nanocrystals in major ampullate gland silk [32,35]. Figure 5a shows the X ray diffraction patterns of flagelliform silk fibers spun by *Argiope trifasciata* spiders with different values of the alignment parameter, α [58]. The correspondence of each value of the alignment parameter with the true stress-true strain curve of a maximum supercontracted flagelliform fiber is also shown in the Figure. 

As expected from the absence of the polyalanine motif in the sequence of flagelliform silk, the diffraction pattern of this fibers do not correspond to β-nanocrystals. In contrast, the unit cell is compatible with a new type of nanocrystals known as polyproline II nanocrystals, that may appear in proteins with a high content of the amino acid glycine [59], consistently with the abundance of –GGX– and –GPG– motifs in flagelliform spidroin [32]. Interestingly, the crystallographic parameters of the unit cell in the a and b directions (mostly perpendicular to the fiber axis) are very similar to those of the β-nanocrystals, so that the only evident distinguishing mark between both types of nanocrystals is the difference in the dimension of the unit cell along the c axis (mostly parallel to the fiber axis). 

Additionally, and in parallel with the previous analyses on major ampullate gland silk, the X ray diffraction data were used to determine the evolution of the crystalline phase upon deformation of the fiber and are summarized in Figure 5b–d. In accordance with the characterization of major ampullate gland silk, the crystallinity increases steadily with increasing values of the alignment parameter. A first significant difference between both types of silk is found in the evolution of the orientation of the nanocrystals, since in flagelliform silk this mechanism sets in only at relatively high values of strain. However, it may be argued that the most contrasting variation between both materials is observed in the evolution of the size of the nanocrystals. In this regard, polyproline II nanocrystals are shown to vary in size and, in particular, an increase is found in the direction parallel to the macroscopic axis of the fiber, as expected from a mechanism by which the crystalline phase is created at the expense of the amorphous one as a result of stretching the fiber.

## 4. The Common Design Principles of Major Ampullate Gland Silks Spun by Entelegyane Spiders

The combination of mechanical and microstructural characterization of spider silk allows proposing a coherent model that accounts for the main observed features of this material. In this context, the differences in the tensile properties of the fibers spun by different species would simply reflect the variation in the quantitative parameters that determine the model. Figure 6 is intended to illustrate this model.

Upon initial loading, the tensile properties of the fiber are determined by a network of hydrogen bonds formed during the spinning of the fiber [60]. The start of the breaking of these hydrogen bonds is marked by the yield stress [50,51], and indicates the end of the elastic regime of the material. From the yield stress, the properties of the fiber are controlled by the elastomeric behavior of the protein chains [22]. However, instead of behaving like a conventional elastomer [61], the mechanical energy is partly dissipated by the irreversible rotation of the β-nanocrystals, that tends to align them with the macroscopic axis of the fiber. This deformation micromechanism seems to be particularly effective in the Orbicularian representatives of the Entelegynae suborder, and might be related with the appearance of the spidroin-2 family of protein and their characteristic –GPG– motif [62,63]. When the β-nanocrystals reach their maximum alignment with the axis of the fiber, a second crystalline phase of polyproline II nanocrystals is formed, most probably by taking the original β-nanocrystals as their nucleation seeds. This way another significant fraction of the mechanical energy imparted to the fiber is dissipated and no longer available to propagate cracks and lead to the breaking of the fiber. It may be argued that the regions of the protein chains that create this second crystalline phase do not sustain significant loads at lower strains and constitute what has been described with the term “hidden length” of the proteins [9,64,65]. This deformation micromechanism is consistent with the necessity of conformational freedom of the chains, for the silk fibers to reach their optimum values of energy to breaking [66] and with the observation of an increase in the size of the microstructural features as observed by atomic force microscopy at high values of strain [67].

## 5. Concluding Remarks

The development of procedures to tailor the tensile properties of individual spider silk fibers, and its combination with single X ray diffraction have contributed to the formulation of a coherent model that explains the relationship between microstructure and mechanical properties in this material. As discussed above it was possible to identify two deformation micromechanisms (three if the initial deformation of the hydrogen bond network is considered): rotation of the β-nanocrystals to increase their alignment with the fiber axis and formation of a second crystalline phase, possibly by polyproline II nanocrystals, that seem to establish a common design pattern of spider silk in Entelegyanae representatives. In this context, quantitative differences between species in the parameters that define both micromechanisms (i.e., the initial angle of the β-nanocrystals with respect to the macroscopic axis of the fiber) would account for the observed variability in their tensile properties.

There are, however, some significant details that will have to be solved before a complete understanding of spider silk can be claimed. To begin with, and despite the accumulation of indirect evidences, the direct observation of polyproline II nanocrystals in major ampullate gland silk is lacking. In addition, it will be necessary to try to establish clear (when possible quantitative) correlations between the mechanical behavior of the fibers, and the ecology and/or genetics of the spinning species. A thorough knowledge of the range of properties exhibit by different spider silks, combined with a detailed molecular characterization of the material, would not only provide a superb perspective of how evolutionary pressures have modelled these outstanding fibers, but it would also be an invaluable tool to guide the rationale design of novel fibers inspired in the natural material.

## Figures and Tables

**Figure 1 molecules-26-01794-f001:**
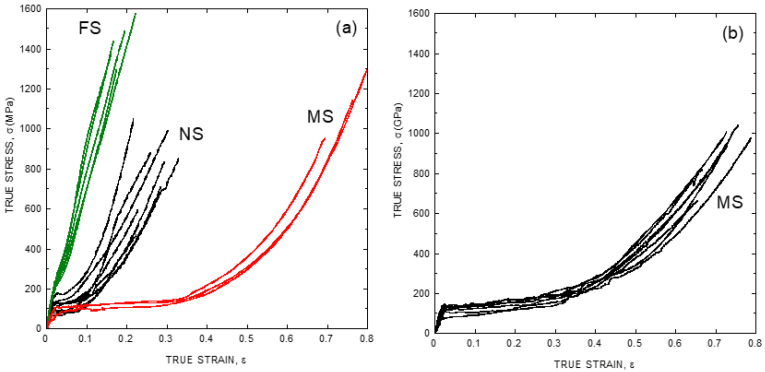
Characteristic tensile properties of spider silk expressed as true stress vs. true strain curves. (**a**) Tensile properties of spider silk spun by *Argiope trifasciata* spiders. FS: forcibly silked, NS: naturally spun, MS: maximum supercontracted. (**b**) Tensile properties of maximum supercontracted (MS) spider silk fibers spun by *Argiope argentata* spiders. The experimental details may be found in the corresponding references by the authors cited in the main text.

**Figure 2 molecules-26-01794-f002:**
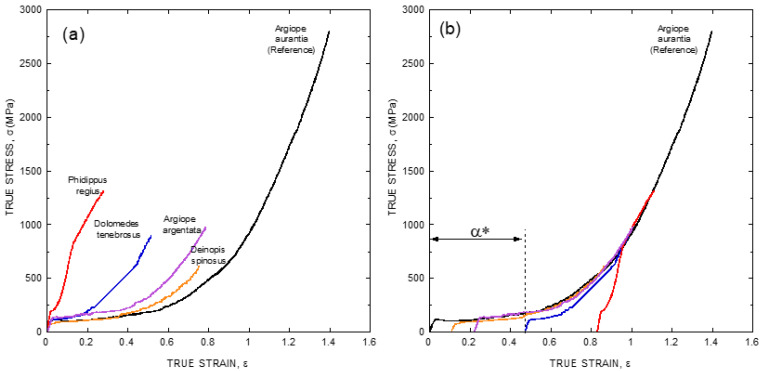
(**a**) Comparison of true stress-true strain curves of fibers spun by different spiders species of the Entelegynae suborder after being subjected to maximum supercontraction (MS curves). (**b**) Definition of the α* parameter by displacing the true stress-true strain curves of the maximum supercontracted fibers along the true strain axis (X-axis) so that it concurs with the *Argiope aurantia* MS curve used as reference at high values of true strain. The experimental details may be found in the corresponding references by the authors cited in the main text.

**Figure 3 molecules-26-01794-f003:**
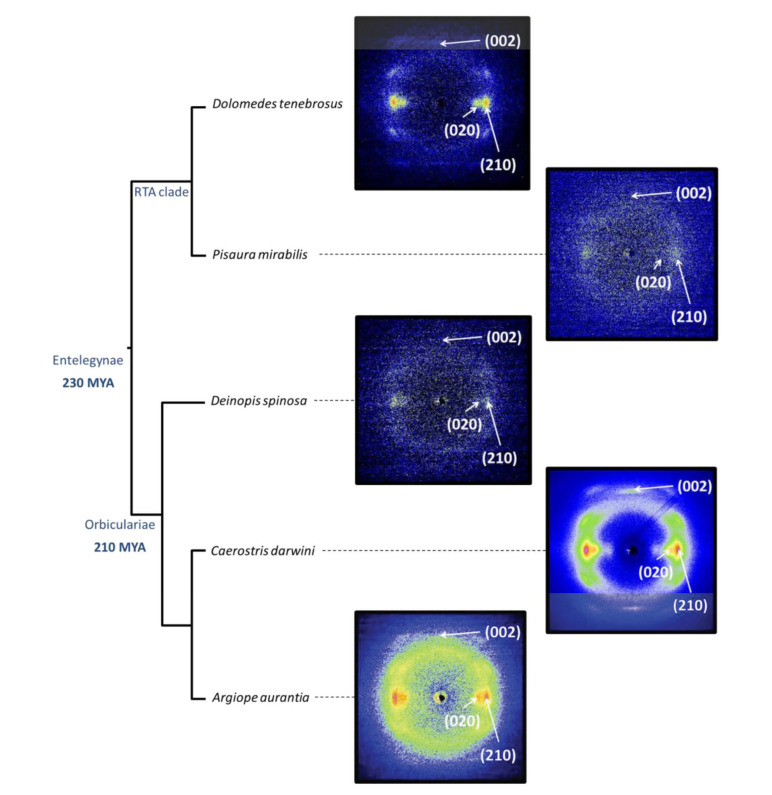
(**a**) X ray diffraction patterns of individual spider silk fibers of selected representatives of the Entelegynae suborder after being subjected to maximum supercontraction. (**b**) Relationship between the extent of supercontraction and the orientation of the nanocrystals with respect to the macroscopic axis of the fiber (measured as the full width at half maximum (FWHM) of the diffraction spot along the azimuthal direction) for the different species. (**c**) Relationship between the extent of supercontraction and the crystallinity of the fibers for the different species. The experimental details may be found in the corresponding references by the authors cited in the main text.

**Figure 4 molecules-26-01794-f004:**
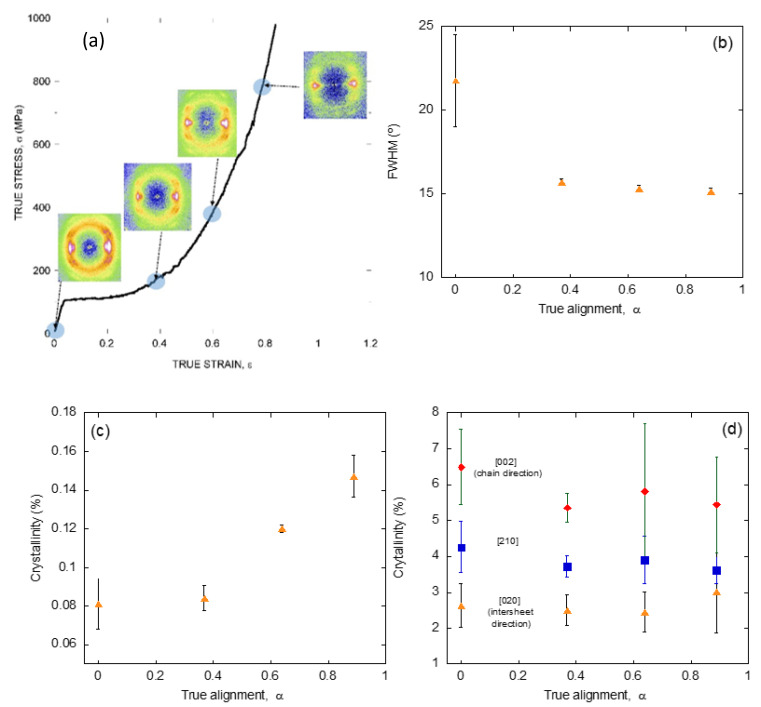
(**a**) X ray diffraction patterns obtained from individual *Argiope trifasciata* spider silk fibers subjected to different values of stretching from the maximum supercontracted state. The approximate value of the corresponding alignment parameter is indicated on an MS curve. (**b**) Evolution of the orientation of the nanocrystals with increasing values of the alignment parameter. (**c**) Evolution of the crystallinity of the fibers with increasing values of the alignment parameter. (**d**) Evolution of the size of the nanocrystals along three independent spatial directions as a function of the alignment parameter. The experimental details may be found in the corresponding references by the authors cited in the main text.

**Figure 5 molecules-26-01794-f005:**
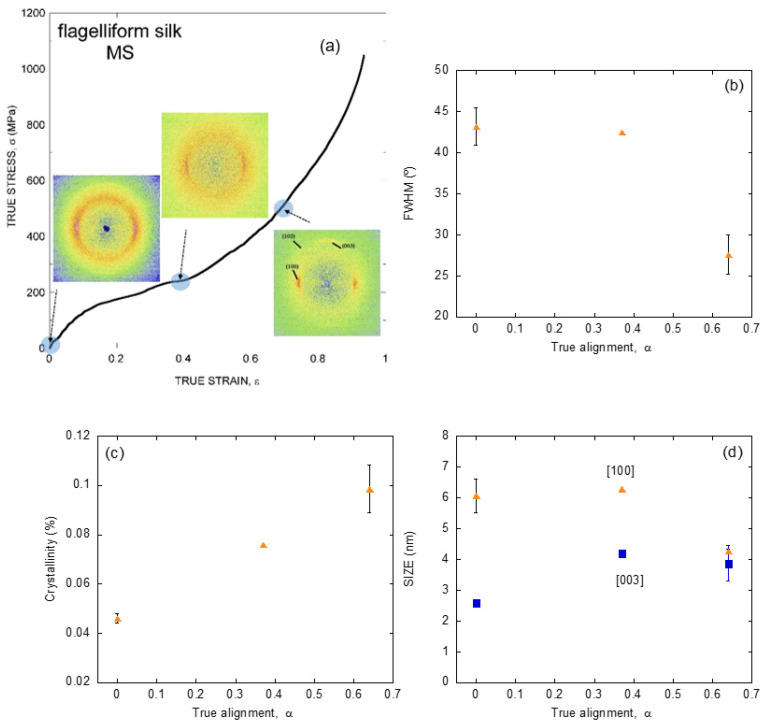
(**a**) X ray diffraction patterns obtained from individual *Argiope trifasciata* flagelliform (or viscid) spider silk fibers subjected to different values of stretching from the maximum supercontracted state. The approximate value of the corresponding alignment parameter is indicated on an MS curve. (**b**) Evolution of the orientation of the nanocrystals with increasing values of the alignment parameter. (**c**) Evolution of the crystallinity of the fibers with increasing values of the alignment parameter. (**d**) Evolution of the size of the nanocrystals as a function of the alignment parameter. The experimental details may be found in the corresponding references by the authors cited in the main text.

**Figure 6 molecules-26-01794-f006:**
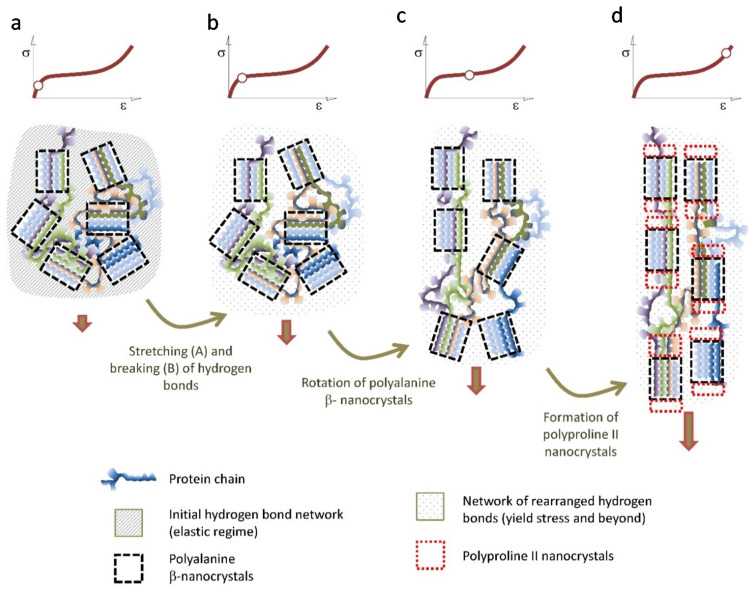
Scheme of the proposed deformation micromechanisms of the Entelegynae major ampullate gland silk fibers. The hydrogen bonds are stretched in the elastic regime (**a**) and begin to break upon reaching the yield stress (**b**). The breaking of the initial network of hydrogen bonds is followed by the rotation of the polyalanine β-nanocrytals that tends to get oriented with the macroscopic axis of the fiber (**c**) When the maximum possible orientation of the polyalanine β-nanocrystals is attained, polyproline II nanocrystals start to form at the ends of the β-nanocrystals (**d**).

## Data Availability

The data presented in this study are available in this article and upon request to the corresponding author.

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
