# Peer review of "Basic Principles in the Design of Spider Silk Fibers"

_molecules, 2021, doi:10.3390/molecules26061794_

Round 1
Reviewer 1 Report
Dear Authors,
I have read your manuscript with attention and interest. The material is consistent and valuable. The submission falls within the scope of the journal and is sufficiently original. However, I have some remarks considering your text, so I suggest publishing it after additional revision. The main attention is to strengthen the content of introduction by presenting key features of chosen material analyzed in the research.
1) It is not clear from your article why spider silk is of specific interest for scientific community. Please add some additional information on spider silk properties. The comparison of spider web mechanical characteristics with other fibers is also worth including in this context. You can cite required information from followed papers:
- Kiseleva A.P., Kiselev G.O., Nikolaeva V.O., Seisenbaeva G., Kessler V., Krivoshapkin P.V., Krivoshapkina E.F. Hybrid spider silk with inorganic nanomaterials // Nanomaterials - 2020, Vol. 10, No. 9, pp. 1853
- Andersson, M.; Johansson, J.; Rising, A. Silk spinning in silkworms and spiders. J. Mol. Sci. 2016, 17, 1290.
2) The authors should specify if the experimental data presented in the article was taken by them or it was collected from other scientific works. If it was done by the authors, please provide the experimental conditions, otherwise please provide literature references.
3) Since one of the main points stated by the authors is that the information given in the article can help the scientific community to obtain artificial silk fibers with superior characteristics, it would be logical to provide some information on current state of research. For instance, the following research Bowen C. H. et al. Recombinant spidroins fully replicate primary mechanical properties of natural spider silk //Biomacromolecules. – 2018. – Т. 19. – â„–. 9. – С. 3853-3860. stated that the mechanical properties of native spider silk were fully replicated in artificial analogues. I suggest you to include the information on particular flaws in artificial silk fibers, which can be improved by the assumptions given in your article.
4) The title of the article «Basic principles in the design of spider silk» sounds rather misleading, since it implies particular guidance how spider silk can be manufactured in nature/by humanity. I suggest the authors to explain their view on this matter in introduction part, providing a context to potential readers.
Best wishes,
Author Response
Please see attachement

Reviewer 2 Report
This manuscript is a review compiling data on spider silk structure correlations to mechanical properties. The major focus is on the a* parameter that this group has defined in previous work. There are several good extensions of their previous analyses that provide interesting information. Still would be nice to see how this fits for minor ampullate silk as it does not supercontract but does show reorientation of the crystalline regions in water. The manuscript is well written (a couple of errors noted below) and the figures are quite good.
Ln 20 should be of not or
Ln 43 needs to be revised
Ln 311 produced not produces
Reviewer 3 Report
This paper overviewing the principals underlying spider silk property variability at various levels is very interesting and timely and well worth publishing in Moecules. I nevertheless have some suggestions for amending parts of the manuscript, primarily as means to produce a more complete picture to that presented thus far.
While the review of the mechanisms driving spider silk properties and the variabilities across and within spider species and other taxonomic groups is reasonably comprehensive, drawing primarily from the author’s own work, it seems disjointed without a very early mention of the role of the different amino acids in driving specific properties. There is mention of this role of amino acids when describing the motifs forming particular structures within section 2. It would makes sense to go over how key amino acids, such as Ala, Gly, Pro are strongly correlated with specific properties. There is an excellent meta-analysis on this by Craig et al. 2020 J Roy. Soc. Interf., in which the correlations of all major amino acids with specific properties across broad spider taxa are analysed. The outcomes of this needs to be mentioned herein as it would tie together the arguments made around a supercontracted “ground state” existing for any particular spider silk that may be predictable from its amino acid composition as the influences of spinning conditions are eliminated.
There are many studies on the inter-individual/specific plasticity of MA silk(and others, e.g. viscous gluey silk and pyriform) properties across diets and environmental conditions (including temperature, humidity, winds, pesticide exposure and so on) by Blamires, Tso, Blackledge, Ayoub, and others. It is worth spending some time embellishing this literature as the molecular mechanisms described in this paper go a long way to explaining the mechanisms driving this phenomena. Some mention of the underlying genes and their variable expression (as in Blamires et al. 2018 Plos One and Miller et al. 2020 Plos One) will tie nicely tie things back to amino acid variability.
Provide reference to support (e.g. Blamires et al. 2017 Annu Rev. Entomol.) to make the point (line 57-59) about major ampullate silk being the most thoroughly analysed of all spider silks, so considered the model for all spider silks.
Some more minor issues are:
Line 43 “..a desired property of the material..”
The figures are all cut in half within the manuscript. Thus I cannot comment, although I am familiar with them through the authors previous work so I’m prepared to accept them. The authors nonetheless should check them and their positioning and appearance in subsequent versions of the manuscript.
